# SFT Memorizes, RL Generalizes:
# A Comparative Study of Foundation Model Post-training

**Tianzhe Chu**♠* **Yuexiang Zhai**♥♠* **Jihan Yang**♦ **Shengbang Tong**♦
**Saining Xie**♠♦ **Dale Schuurmans**♠♣ **Quoc V. Le**♠ **Sergey Levine**♥ **Yi Ma**♠♥

## Abstract

Supervised fine-tuning (SFT) and reinforcement learning (RL) are widely used post-training techniques for foundation models. However, their respective role in enhancing model generalization in rule-based reasoning tasks remains unclear. This paper studies the comparative effect of SFT and RL on generalization and memorization, focusing on text-based and visual reasoning tasks. We introduce `GeneralPoints`, an arithmetic reasoning card game, and also consider `V-IRL`, a real-world navigation environment, to assess how models trained with SFT and RL generalize to unseen variants in both novel textual rules and visual domains. We show that RL, especially when trained with an outcome-based reward, generalizes in both the rule-based textual and visual environments. SFT, in contrast, tends to memorize the training data and struggles to generalize out-of-distribution in either scenario. Further analysis reveals that RL improves the model's underlying visual recognition capabilities, contributing to its enhanced generalization in visual domains. Despite RL's superior generalization, we show that SFT is still helpful for effective RL training: SFT stabilizes the model's output format, enabling subsequent RL to achieve its performance gains. These findings demonstrate the advantage of RL for acquiring generalizable knowledge in complex, multimodal tasks.

*Equal contribution . ♠HKU, ♥UC Berkeley, ♣Google DeepMind, ♦NYU, ♣University of Alberta. All experiments are conducted outside of Google. Project page: https://tianzhechu.com/SFTvsRL. Correspondence to: Tianzhe Chu <tianzhechu@gmail.com>, Yuexiang Zhai <simonzhai@berkeley.edu>.

*Proceedings of the 42ⁿᵈ International Conference on Machine Learning*, Vancouver, Canada. PMLR 267, 2025. Copyright 2025 by the author(s).

## 1. Introduction

Although SFT and RL are both widely used for foundation model training (OpenAI, 2023b; Google, 2023; Jaech et al., 2024; DeepSeekAI et al., 2025), their distinct effects on *generalization* (Bousquet & Elisseeff, 2000; Zhang et al., 2021) remain unclear, making it challenging to build reliable and robust AI systems. A key challenge in analyzing the generalizability of foundation models (Bommasani et al., 2021; Brown et al., 2020) is to separate data memorization[1] from the acquisition of transferable principles. Thus, we investigate the key question whether SFT or RL primarily memorize training data (Allen-Zhu & Li, 2023a; Ye et al., 2024; Kang et al., 2024), or whether they learn generalizable rules that can adapt to novel task variants.

To address this question, we focus on two aspects of generalization: textual rule-based generalization and visual generalization. For textual rules, we study the ability of a model to apply learned rules (given text instructions) to variants of these rules (Zhu et al., 2023; Yao et al., 2024; Ye et al., 2024). For vision-language models (VLMs), visual generalization measures the consistency of performance with variations in visual input, such as color and spatial layout, within a given task. For studying text-based and visual generalization, we investigate two different tasks that embody rule-based and visual variants. Our first task is `GeneralPoints`, an original card game task similar to `Points24` of RL4VLM (Zhai et al., 2024a), which is designed to evaluate a model's *arithmetic reasoning capabilities*. The model receives four cards (presented as a text description or an image), and is required to compute a target number (24 by default) using each card's numerical value exactly once. Second, we adopt `V-IRL` (Yang et al., 2024a), a real-world navigation task that focuses on the model's *spatial reasoning capabilities*.

We adopt a multi-step RL framework similar to Zhai et al. (2024a), by instantiating RL after running SFT on

---

[1]We use "memorization" the refer a model's capacity to generate near-exact copies of training examples when prompted based on information present in the training dataset. This definition explicitly excludes bitwise or codewise replication of training data within the model itself.

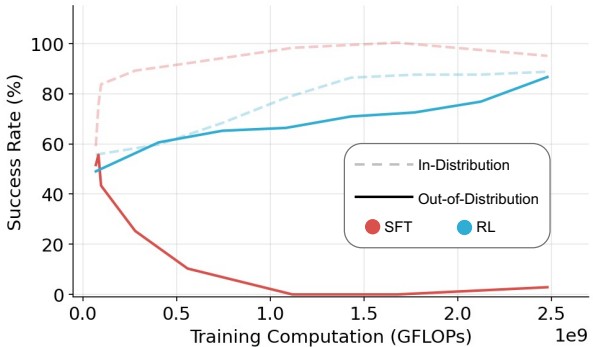

Figure 1: **A comparative study of RL and SFT on the visual navigation environment** `V-IRL` **(Yang et al., 2024a) for OOD generalization.** OOD curves represent performance on the same task, using *a different textual action space*. See detailed descriptions of the task in Section 5.1.

the backbone model (Dubey et al., 2024), using the sequential revision formulation (Snell et al., 2024). In both `GeneralPoints` and `V-IRL`, we observe that RL learns generalizable rules (expressed in text), where in-distribution performance gains also transfer to unseen rules. In contrast, SFT appears to memorize the training rules and does not generalize (see Figure 1 for an example). Beyond textual rule-based generalization, we further investigate generalization in the visual domain and observe that RL also generalizes to visual OOD tasks, whereas SFT continues to struggle. As a by-product of the visual OOD generalization capability, our multi-turn RL approach **achieves state-of-the-art performance** on the `V-IRL` mini benchmark, by **+33.8%** (44.0%→77.8%) (Yang et al., 2024a), highlighting the generalization capability of RL. To understand how RL affects the visual abilities of a model, we conducted additional analysis on `GeneralPoints`, revealing that training RL with an outcome-based reward function (Cobbe et al., 2021) improves visual recognition capabilities. Although RL exhibits superior generalization compared to SFT, we show that SFT is still necessary to stabilize the model's output format, enabling RL to achieve its performance gains. Last but not least, we observe that scaling up the inference time compute by increasing the number of maximal steps leads to better generalization.

## 2. Related Works

**Post-training.** Post-training is crucial for enhancing model performance (Zhang et al., 2022; Hoffmann et al., 2023; OpenAI, 2023b; Google, 2023; Touvron et al., 2023). This stage commonly utilizes large-scale supervised fine-tuning (SFT) (Radford et al., 2018; Brown et al., 2020; Radford et al., 2021; Wei et al., 2022a; Chung et al., 2022; Zhou et al., 2024a) and/or reinforcement learning (RL) (Ziegler et al., 2019; Ouyang et al., 2022; Sun et al., 2024; Abdulhai et al., 2023; Zhou et al., 2024b; Zhai et al.,

2024a). SFT adapts pre-trained models to downstream tasks by training them on task-specific, often instruction-formatted datasets. Previous work, such as FLAN (Wei et al., 2022a), demonstrates that fine-tuning on diverse instruction-tuning datasets significantly enhances zero-shot performance on unseen tasks. Furthermore, LIMA (Zhou et al., 2024a) shows that supervised fine-tuning acts as a "format teacher" effectively adapting the model's responses to a desired format while leveraging the capabilities of pre-trained LLMs. In contrast, RL (Ziegler et al., 2019; Ouyang et al., 2022; Sun et al., 2024; Ramamurthy et al., 2023; Abdulhai et al., 2023; Zhou et al., 2024b; Zhai et al., 2024a) has been primarily used to align models with human preferences or training the foundational model to solve a specific task (Abdulhai et al., 2023; Zhou et al., 2024b; Zhai et al., 2024a; Chen et al., 2024b). Our work differs from prior studies, as we aim to comparatively analyze the generalization and memorization of SFT and RL on *both LLM and VLM*, while previous studies have focused primarily on only one of these two post-training methods (or only study LLM or VLM) or on only one post-training method.

**Memorization and generalization in LLM/VLM.** Several studies have examined the interplay between memorization and generalization in neural networks (Han et al., 2022; Carlini et al., 2022; Yang et al., 2023). In LLMs, memorization can manifest as the model memorizing the training data (Carlini et al., 2022; Jiang et al., 2024; Kang et al., 2024), while generalization reflects the divergence between the model's output distribution and the pre-training data distribution (Zhang et al., 2023). Prior studies suggest that LLMs exhibit more overfitting on simpler, knowledge-intensive tasks and greater generalization on more complex, reasoning-intensive ones (Wang et al., 2024; Qi et al., 2024). For example, recent studies (Ye et al., 2024; Allen-Zhu, 2024; Allen-Zhu & Li, 2023a;b; 2024; Tong et al., 2024b) have demonstrated that LLMs develop reasoning skill sets beyond their training data by pre-computing reasoning graphs before autoregressive generation, which provides compelling evidence of generalization. Our study takes a different approach by investigating the role of different post-training paradigms on memorization versus generalization in the context of textual ruled-based and visual variants. We conduct comparative studies in both unimodal (LLM) and multimodal (VLM) settings, and demonstrate that RL leads to better generalization performance than SFT.

**Scaling up inference-time compute.** Recent research has increasingly focused on scaling up inference-time computation to improve model performance (Wei et al., 2022b; Yao et al., 2024; Snell et al., 2024; Jaech et al., 2024). Early studies (Wei et al., 2022b; Yao et al., 2024) prompted

models to generate intermediate reasoning steps and extend the responses before producing a final answer. Subsequent work (Zelikman et al., 2022; Feng et al., 2023; Tian et al., 2024; Chen et al., 2024a; Snell et al., 2024) has demonstrated that fine-tuning verifiers during inference improves model accuracy, effectively utilizing test-time computation. Notably, recent findings (Jaech et al., 2024; DeepSeekAI et al., 2025) reveal "scaling laws" for inference-time compute, highlighting significant performance gains with increased computational resources. Our work builds upon these findings in two ways. First, we integrate insights from inference-time verification into a multi-turn RL formulation that allows the model to identify and correct its errors. Second, we examine the impact of inference-time verification on RL generalization, demonstrating that scaling up inference-time verification (in terms of the maximum number of verification steps) is a key for RL to generalize.

**Improving visual capability in VLMs.** While VLMs have demonstrated remarkable skill across a wide range of challenging tasks, such as solving advanced college exam questions (Lu et al., 2023; Yue et al., 2024a;b) and spatial understanding tasks (Yang et al., 2024a;b), they also exhibit limitations in visual perception (Zhai et al., 2024a;b; Tong et al., 2024c;d; Rahmanzadehgervi et al., 2024). Prior efforts to enhance VLMs' visual perception include combining multiple visual encoders (Tong et al., 2024d; Kar et al., 2025; Tong et al., 2024a), curating high-quality SFT data (Chen et al., 2023; Liu et al., 2024; Tong et al., 2024a), and improving the SFT training recipe by unfreezing the visual backbone (Liu et al., 2023; Tong et al., 2024a). While these prior works primarily focus on experiments during the SFT stage, our work demonstrates that RL can also improve visual perception.

## 3. Preliminaries

**Standard RL terminology.** We consider finite horizon decision making, and adopt standard notation from the classical RL literature (Sutton & Barto, 2018; Agarwal et al., 2019), where $\mathcal{S}$ denotes the state space, $\mathcal{A}$ denotes the action space, $r : \mathcal{S} \times \mathcal{A} \to \mathbb{R}$ denotes the reward function, and $T$ denotes the maximum number of steps per episode. The goal is to learn a policy $\pi : \mathcal{S} \to \mathcal{A}$ that maximizes the overall return $\max_{\pi \in \Pi} \mathbb{E}_\pi \left[ \sum_{t=0}^{T} r_t \right]$, where $r_t$ denotes $r(s_t, a_t)$. Without loss of generality, we use $\pi(a|s) \in [0, 1]$ to denote probability of $\pi$ choosing $a$ at $s$.

**Adapting RL terminology to LLM/VLM with a verifier.** We adopt a multi-turn RL setting for foundation model training (Zhai et al., 2024a). Let $\mathcal{V}$ represent the discrete and finite vocabulary (token) space. The input and output text spaces are denoted by $\mathcal{V}^m$ and $\mathcal{V}^n$ respectively, where

$m$ and $n$ are the maximum token length of the input sequence $\boldsymbol{v}^{\text{in}}$ and output sequence $\boldsymbol{v}^{\text{out}}$. For models requiring visual inputs (VLM), we define $\mathcal{O}$ as the space of all RGB images. The state space, denoted by $\mathcal{S}$, is defined as $\mathcal{S} := \mathcal{V}^m \times \mathcal{O}$ for VLM, and $\mathcal{S} := \mathcal{V}^m$ for LLM. The action space $\mathcal{A}$ is defined as $\mathcal{A} := \mathcal{V}^n$. We use $\text{VER} : \mathcal{V}^n \to \mathbb{R} \times \mathcal{V}^k$ to denote a verifier, which evaluates the outcome of $\boldsymbol{v}^{\text{out}}$ and generates an outcome-based reward function (Cobbe et al., 2021; Hosseini et al., 2024; Snell et al., 2024; Setlur et al., 2024) $r$ along with textual information $\boldsymbol{v}^{\text{ver}}$. Mathematically, at time $t$, $\text{VER}(\boldsymbol{v}_t^{\text{out}}) \mapsto (r_t, \boldsymbol{v}_t^{\text{ver}})$. Similar to Zhai et al. (2024a), we treat the model with parameter $\theta$ as our policy network $\pi_\theta : \mathcal{S} \to \mathcal{V}^n$, and adopt PPO (Schulman et al., 2017) as the backbone RL algorithm for updating $\pi_\theta$.

**Sequential revision.** For modeling the state-action transition, we adopt the sequential revision formulation (Snell et al., 2024). Specifically, at time step $t = 0$ the initial input $\boldsymbol{v}_0^{\text{in}}$ consists of the system prompt. For subsequent time steps ($t \geq 1$), the input prompt $\boldsymbol{v}_t^{\text{in}}$ comprises the system prompt concatenated with all prior model and verifier outputs, denoted by $[\boldsymbol{v}_k^{\text{out}}, \boldsymbol{v}_k^{\text{ver}}]_{k=0}^{t-1}$. An illustration of the sequential revision is provided in Figure 2 (also see Figure 5 of Snell et al. (2024)), and an example of the state-action transition is shown in Figure 3.

## 4. Evaluation Tasks

To evaluate the generalization of different post-training methods, we select two tasks that each offer *rule* and *visual variations*. The first task, GeneralPoints, is a new environment we have designed that allows assessment of arithmetic reasoning abilities (Section 4.1). The second task, V-IRL (Yang et al., 2024a), is chosen to examine the model's reasoning capabilities in an open-world visual navigation domain (Section 4.2).

### 4.1. The General Points Environment

Our original GeneralPoints environment, instantiated on top of the Points24 environment (Zhai et al., 2024a), is designed to evaluate generalization of arithmetic reasoning. Each state $s$ of the environment contains 4 cards, described as text (in the GP-L variant) or presented as an image (in the GP-VL variant); see Figure 2 left for a visual example of GeneralPoints. The goal is to *produce an equation that equals a target number* (24 by default) using all 4 numbers from the cards *exactly once*. Detailed examples of the state-action transitions are provided in Appendix A.2. Note that when input from GeneralPoints is presented in an image (GP-VL), it naturally introduces additional visual challenges requiring the VLM to *recognize all cards before solving the equation*.

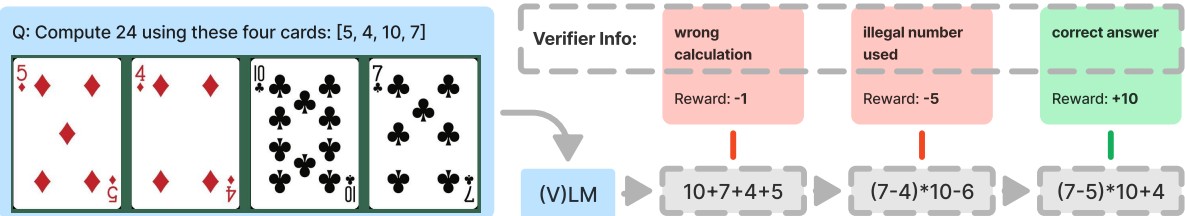

**Figure 2:** **An example of the sequential revision formulation with a verifier.** The model generate the next answer $v_{t+1}^{\text{out}}$ conditioned on *all previous answers and information* $(v_i^{\text{out}}, v_t^{\text{ver}}, 0 \leq i \leq t)$ from the verifier.

---

**System Prompt ($v_0^{\text{in}}$)**

[Task Description] You are an expert in {task name}, you are observing {purely language/vision-language inputs + <image>}. You are currently at {state related info}. Please follow {tasks rules}.

[Output] Your response should be a valid json file in the following format:
{task related information and answer}

---------------------------------------------------------------------------------------------------------------------------

**Appending previous model and verifier outputs to obtain $v_t^{\text{in}}$**

$v_t^{\text{in}} = [v_0^{\text{out}}, v_0^{\text{ver}}, v_1^{\text{out}}, v_1^{\text{ver}}, \ldots, v_{t-1}^{\text{out}}, v_{t-1}^{\text{ver}}]$         $\triangleright\; v_t^{\text{in}} = \text{concat}\left(v_0^{\text{in}}, [v_k^{\text{out}}, v_k^{\text{ver}}]_{k=0}^{t-1}\right)$

---

**Model output ($v_t^{\text{out}}$) and Verifier Output ($v_t^{\text{ver}}$)**

{Task related json outputs}, {You success/fail}.         $\triangleright\; v_{t+1}^{\text{in}} = \text{concat}(v_t^{\text{in}}, v_t^{\text{out}}, v_t^{\text{ver}})$

**Figure 3:** **An template of our prompt update for constructing $v_{t+1}^{\text{in}}$.** The brown parts marks the task and related information, and the purple parts denote the state ($s_t$) specific info. The blue and red describe the output from the model and verifier, respectively.

**Rule variations.** To study whether the model learns arithmetic operations or simply memorizes the post-training data, we introduce rule variations in GeneralPoints. These variations consist of interpreting the symbols 'J', 'Q', and 'K' either as '11', '12', and '13', respectively, or all as the same number '10'. These variations ensure a rigorous evaluation of the model's ability to generalize arithmetic reasoning across diverse settings. Each rule is specified as text in the input prompt, see the {tasks rules} part in Figure 3. For studying ruled based generalization, we post-train the model *using one rule*, then *evaluate using a different rule*.

**Visual variations.** The GeneralPoints environment can also be naturally customized to evaluate generalization across visual variants. Since the major visual challenge is to *recognize the number of each card*, agnostic to the *color* of the cards, we consider the cards with different colors as visual variants of the task. In the visual generalization setting, we train the model using cards of one color, then test OOD performance using the other color.

**4.2. The V-IRL Environment**

While the GeneralPoints environment is designed to assess arithmetic reasoning abilities, we further utilize the V-IRL environment (Yang et al., 2024a) to study *spatial reasoning* ability in an open-world navigation domain that

uses *realistic visual input*. As in GeneralPoints we consider two versions of the environment, one (V-IRL-L) that consists of pure language descriptions,[2] and another (V-IRL-VL) that includes vision-language input. The major visual challenge in V-IRL involves *recognizing different landmarks from the visual observation*[3] before taking an action. The goal is to *navigate to a target location by following a set of instructions that contain spatial information*. A detailed example of one environment step is shown in Appendix B.2.

**Rule variations.** To evaluate whether the model possesses spatial knowledge or simply memorizes post-training data, we consider two distinct action space configurations. The first variant utilizes an *absolute orientation* action space, which includes {'north', 'northeast', 'east', 'southeast', 'south', 'southwest', 'west', 'northwest'}. The second variant employs a *relative orientation* action space, containing {'left', 'right', 'slightly left', 'slightly right'}. This relative configuration adjusts the current orientation by 90 degrees or 45 degrees to the left or right, respectively. An

---

[2]The visual input can be parsed into pure text description, see more details in Yang et al. (2024a) and an illustration of pure text the version in Figure 14.

[3]See Figure 4, the model needs to recognize landmarks like **The Dutch**, **Lola Taverna**, and **Shuka** from the visual observation, and relate these landmarks with the textual instructions for taking the right action.

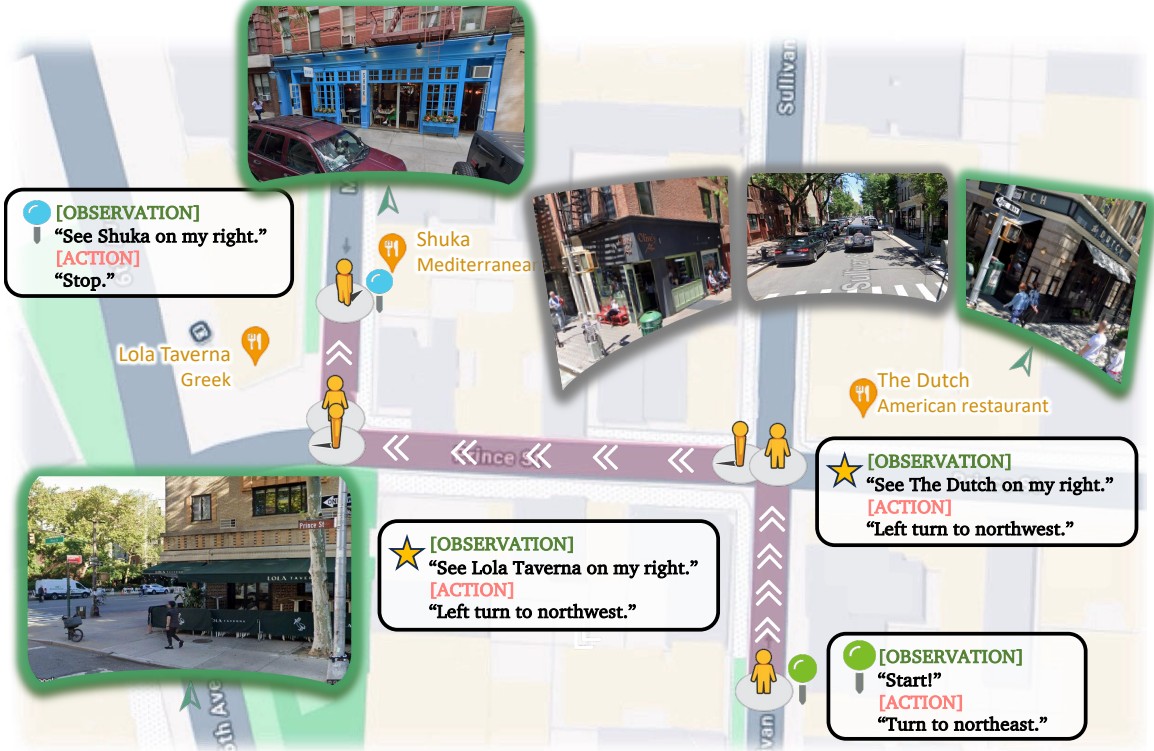

[OBSERVATION]
"See Shuka on my right."
[ACTION]
"Stop."

Shuka
Mediterranean

Lola Taverna
Greek

The Dutch
American restaurant

[OBSERVATION]
"See The Dutch on my right."
[ACTION]
"Left turn to northwest."

[OBSERVATION]
"See Lola Taverna on my right."
[ACTION]
"Left turn to northwest."

[OBSERVATION]
"Start!"
[ACTION]
"Turn to northeast."

⭐ First, **turn slightly right** towards the northeast and walk a short distance until you reach the next intersection, where you'll see **The Dutch** on your right. Next, make a **sharp left turn** to head northwest. Continue for a while until you reach the next intersection, where **Lola Taverna** will be on your right. Finally, **turn slightly right** to face northeast and walk a short distance until you reach your destination, **Shuka**, which will be on your right.

Figure 4: **Demonstration of one navigation task in** `V-IRL`. Agent navigates from place to place following the given linguistic navigation instructions in `V-IRL`. The navigation procedure is shown at the top, with the navigation instructions displayed below. Visual observation-related information is highlighted in green, while action-related information is marked in orange.

overview of a navigation task in `V-IRL` is provided in Figure 4, and a detailed state-action transition in `V-IRL` is provided in Figure 13 (in Appendix B.2).

**Visual variations.** The key visual challenge in `V-IRL` is to recognize landmarks from the visual observations (e.g., the green parts in Figure 4). Since the `V-IRL` environment contains visual observations from different cities, we can assess visual generalization in `V-IRL` by training the model to navigate in one location and then evaluate its performance in *different locations*.

## 5. Results

In this section, we present experiments that investigate the generalization abilities induced by post-training with RL and SFT. We adopt Llama-3.2-Vision-11B (Dubey et al., 2024) as the backbone model. Following the standard pipelines of RLHF (Ouyang et al., 2022) and RL4VLM (Zhai et al., 2024a), we initialize the model with SFT before running RL. We specifically study the follow-

ing questions. Section 5.1: how does SFT or RL affect the model's generalization to different rules? Section 5.2: when the model contains a visual component, how does RL/SFT affect its generalization to different visual variants? Section 5.3: how does RL/SFT affect visual recognition capability in a VLM? Section 5.4: what role does SFT play in RL training? Section 5.5: how does the number of verification iterations affect generalization?

### 5.1. Generalization across Rules

We evaluate the performance of different post-training methods on `GeneralPoints` and `V-IRL`, each of which has a pure language (`-L`) and a vision-language (`-VL`) variant, and each encompassing rule variations. For each task, we separately scale the training compute for RL and SFT on a single rule. We consider the results on the trained rule as in-distribution (ID) performance, whereas results on the *unseen* rules measures out-of-distribution (OOD) generalization. In `GeneralPoints`, the ID case treats all 'J', 'Q', 'K' as 10, and the OOD cases interprets them as 11, 12, and 13. As for `V-IRL`, the ID case adopts the *absolute orienta-*

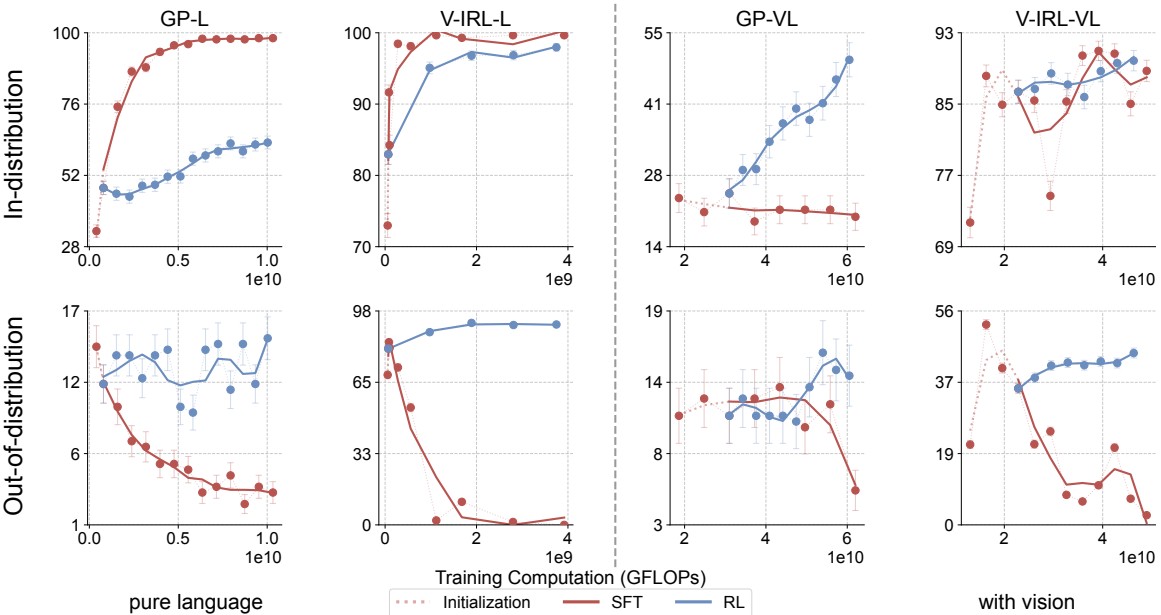

Figure 5: **Success rate (%) - GFLOPs trendlines for RL and SFT on `GeneralPoints` and `V-IRL`.** The top row shows in-distribution performance, while the bottom row shows out-of-distribution performance. Results are presented for both pure language (`-L`) and vision-language (`-VL`) variants of each task. For `GeneralPoints`, we report the episode success rate, while for `V-IRL`, we report per-step accuracy with overall success rate in Figures 1 and 18. Detailed evaluation setups (and curve smoothing) are provided in Appendix C.3.

*tion* coordinate system and the OOD case uses the *relative orientation* action space. Other details and additional experimental setup can be found in Appendix C.

**RL generalizes, SFT memorizes.** As illustrated in Figure 5, RL consistently improves OOD performance on all tasks, including both unimodal (LLM) and multimodal (VLM). Specifically, Figure 6 demonstrates that RL achieves an increase of **+3.5%** on GP-L (11.5% → 15.0%) and **+11.0%** on V-IRL-L (80.8% → 91.8%). Even with the additional challenge of visual recognition in the VLM, RL maintains consistent performance improvements of **+3.0%** (11.2% → 14.2%) on GP-VL and **+9.3%** (35.7% → 45.0%) on V-IRL-VL, respectively. In contrast, SFT consistently exhibits performance degradation across all OOD evaluations on all tasks: **-8.1%** on GP-L (11.5% → 3.4%), **-79.5%** on V-IRL-L (80.8% → 1.3%), **-5.6%** (11.2% → 5.6%) on GP-VL, and **-33.2%** (35.7% → 2.5%) on V-IRL-VL.

### 5.2. Generalization in Visual Out-of-Distribution Tasks

Section 5.1 demonstrates that RL yields generalization across rule variations, whereas SFT exhibits the opposite trend. Since VLMs also incorporate a visual modality, we next study the effects of *visual* variation in OOD generalization. For `GeneralPoints`, we train the VLM using the black suits (♠, ♣) and test out-of-distribution perfor-

mance on the red suits (♥, ♦). For `V-IRL`, we train the model on routes collected in New York City and evaluate it on the original `V-IRL` VLN mini benchmark (Yang et al., 2024a) containing routes from *various cities worldwide* (see Appendix B.1 for details). Note that the rules remain consistent across experiments in this section.

**RL generalizes in visual OOD tasks.** As shown in Figure 7, we observe that RL still *generalizes in visual OOD tasks*, while SFT continues to suffer. Specifically, in GP-VL and VIRL-VL, RL achieves performance improvements of **+17.6%** (23.6% → 41.2%), **+61.1%** (16.7% → 77.8%), whereas SFT suffers from performance decreases of **-9.9%** (23.6% → 13.7%) and **-5.6%** (16.7% → 11.1%). As a byproduct of this visual OOD study, we also show that our multi-turn RL formulation **improves the state-of-the-art results** (see Table 5 of Yang et al. (2024a)) on the V-IRL mini benchmark by **+33.8%** (44.0% → 77.8%). Notably, unlike the previous state-of-the-art approach reported in V-IRL, which relies on a two stage VLM-LLM collaboration technique and tailored prompt engineering on closed-sourced model (OpenAI, 2023a), our end-to-end RL approach enables an open-sourced model (Dubey et al., 2024) to reach superior performance.

### 5.3. RL Improves Visual Capabilities

Building upon the above observation that VLMs trained with RL generalize to visual OOD tasks (Section 5.2), we

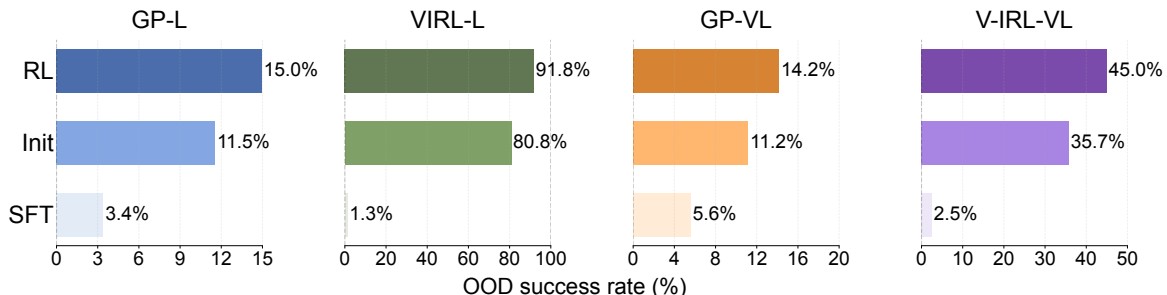

Figure 6: **Comparison of out-of-distribution performance under rule variants.** We report the success rate for `GeneralPoints` and **per-step-accuracy** for `V-IRL`. For each subplot, RL and SFT are trained with equal computation, and their shared initial checkpoint (marked as Init) is set as baseline. Detailed setups are provided in Appendix C.3.

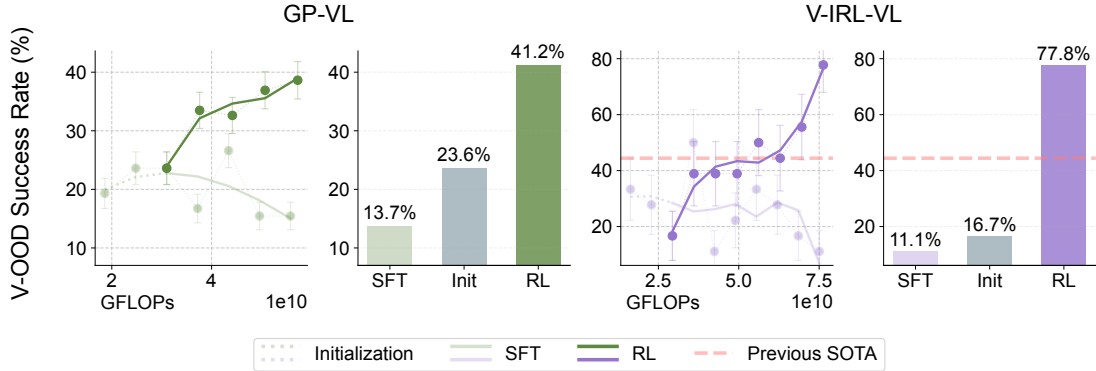

Figure 7: **Comparison of out-of-distribution performance under visual variants.** Similar to Figures 5 and 6, we present both the performance dynamics (shown as lines) and final performance (shown as bars) for visual out-of-distribution evaluations. The previous state-of-the-art on `V-IRL` VLN mini benchmark (Yang et al., 2024a) is marked in orange. Detailed evaluation setups (and curve smoothing) are provided in Appendix C.3.

consider a natural follow-up question: *How does RL affect VLMs' visual capabilities*? To study this question, we conducted additional ablation studies in the `GP-VL` environment to investigate the OOD performance of RL and SFT, along with the model's visual recognition accuracy, in terms of recognizing the 4 cards from the input image. In particular, we study how scaling post-training compute via RL/SFT both affects generalization in rule-based OOD (Figure 8 left), and visual recognition accuracy and visual OOD (Figure 8 right).

**Scaling RL improves visual recognition accuracy in VLM training.** As shown in Figure 8, we observe that the VLM's visual recognition accuracy largely affects the overall performance, which was similarly observed in Zhong et al. (2024). In addition, scaling up RL compute also improves visual recognition accuracy, as a byproduct of its generalization capability, while scaling SFT deteriorates both visual recognition accuracy and overall performance. Additional experimental results are provided in Figures 16 and 17 of Appendix D.1.

### 5.4. The Role of SFT for RL Training

Despite the superiority of RL in generalizing the model's reasoning and visual capabilities, as discussed previously, the experimental pipeline still instantiates RL *after SFT*. In this subsection, we focus on another key question: *Is SFT necessary for RL training*? To answer this question, we conduct additional experiments that directly apply end-to-end RL to post-train the base model Llama3.2 using `GeneralPoints` in the purely language case (Figure 9).

**SFT is necessary for RL training when the backbone model does not follow instructions.** Figure 9 shows that without SFT, all end-to-end RL runs fail to improve. More specifically, we observe that without SFT, the base model suffers from poor instruction following capability. A detailed failure case is provided in Figure 20 (in Appendix D.3), revealing that the base Llama-3.2-Vision-11B model tends to generate long, tangential, and unstructured responses. This issue makes it impossible to retrieve task-related information and rewards for RL training. Note that due to the difference in backbone model, our results do not contradict with DeepSeekAI et al. (2025), which suggests that SFT is unnecessary for downstream RL training.

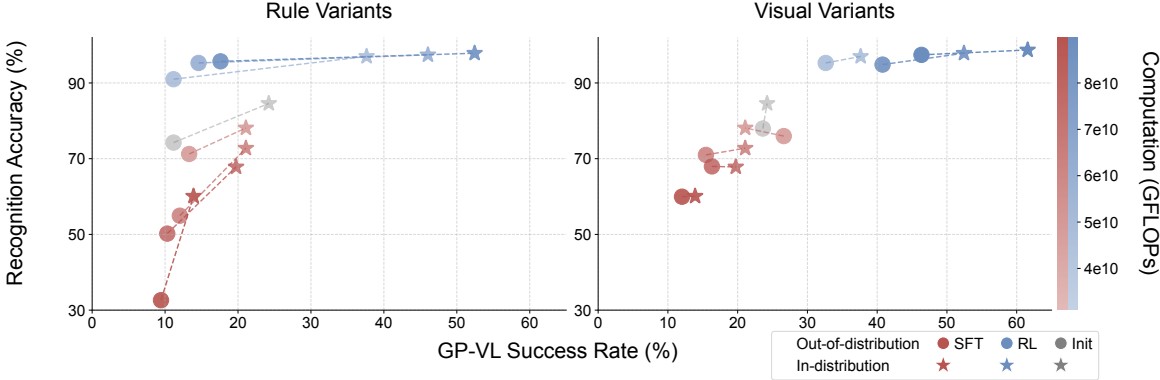

Figure 8: **Recognition vs. success rate for RL and SFT under different variants in `GP-VL`.** We report both in-distribution (red) and OOD (blue) performance of recognition (y-axis) and episode success rate (x-axis). We denote the training compute of each data point via transparency (color bar) while connected (★-○) pairs are evaluated using same checkpoints. As scaling up post-training compute, RL improves both recognition and overall accuracy, while SFT shows opposite effect.

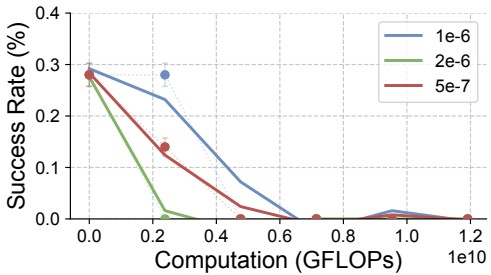

Figure 9: **RL experiments on `GP-L` without SFT initialization.** All trials fail due to poor instruction following capability of the base model.

### 5.5. Role of Verification Iterations

Verification serves as another crucial component in our multi-step training and evaluation pipeline (see Figures 2 and 3). To validate its necessity and better understand its effect, we conduct RL experiments with different verification iterations $\{1, 3, 5, 10\}$ using GP-L (Figure 10).

**Scaling up verification improves generalization.** In Figure 10, we observe that RL generalizes better with more verification steps. More specifically, under the same computational budget across all experiments, we observe improvements of **+2.15%** (3 steps), **+2.99%** (5 steps), **+5.99%** (10 steps). In contrast, in the case with one verification step, we only observe a marginal improvement of **+0.48%** in OOD performance improvement.

## 6. Conclusion, Discussion, and Limitations

In this paper, we present a comprehensive analysis of the generalization effects of foundation model post-training techniques, specifically RL and SFT. Through extensive

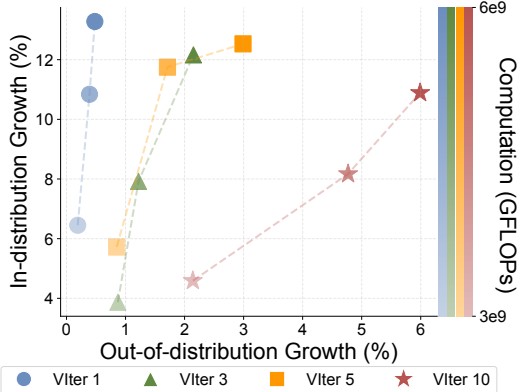

Figure 10: **In-distribution vs. OOD performance growth on `GP-L`.** We record RL experiments with different number of verification iterations (VIter) as scaling up training compute (color transparency).

experiments on the `GeneralPoints` and `V-IRL` tasks, we demonstrated that RL exhibits superior performance in learning generalizable knowledge, while SFT tends to merely memorize the training data, across both the rule and visual variations. This phenomenon consistently occurs across multimodal arithmetic and spatial reasoning capabilities. In addition, we studied the effect of RL on visual recognition, the role of SFT, and the role of verification steps. During our study, two challenges were not resolved.

**Failure of SFT on `GP-VL`.** In Figure 5 for `GP-VL`, we observe that SFT fails to achieve a comparable in-distribution performance with RL. To mitigate the variance introduced by hyperparameter choices, we additionally conduct 10 more experiments with different learning rates and tunable components (Figure 16), none of which exhibits a strong increasing trend like RL (Figure 17).

Given our observation that scaling up SFT degrades visual recognition capabilities (Figure 8), we hypothesize that SFT locally overfits to reasoning tokens while neglecting recognition tokens, possibly due to the higher frequency of reasoning tokens (see Figure 11 as example). We leave further investigation to future work.

**Limits of RL in corner cases.** As discussed in Section 5.4, SFT is necessary for effective RL training on Llama-3.2. We investigate applying RL to an overly-tuned SFT checkpoint. As demonstrated in Figure 19, RL is unable to recover out-of-distribution performance when starting from such a checkpoint. Example failure cases are illustrated in Figure 21, where the model collapses to the training rule. These results, together with findings in Section 5.4, indicate that RL has limited effectiveness when applied to extremely underfit or overfit initial checkpoints. Further research is needed to delineate the conditions under which SFT facilitates effective RL.

## Impact Statement

This paper presents work aimed at advancing the field of Machine Learning. While the study includes tasks such as `GeneralPoints`, which is a synthetic environment, and `V-IRL`, a real-world map simulator, our work is confined to controlled research settings. The `V-IRL` environment is designed as a simulated proxy for real-world tasks, but no deployment or interaction with actual real-world systems or data was involved. The methods, environments, and tasks investigated in this study were constructed to advance our understanding of model generalization without introducing any foreseeable societal or ethical implications.

## Acknowledgements

YZ would like to thank Xiaoxuan Feng for beautifying Figure 4. We would like to thank Jincheng Mei and Doina Precup for feedbacks on earlier manuscripts. Yi Ma would like to acknowledge support from the joint Simons Foundation-NSF DMS grant #2031899, the ONR grant N00014-22-1-2102, the NSF grant #2402951, and also support from and the HKU startup, the Hong Kong Center for Construction Robotics Limited (HKCRC) Award 052245, and JC Club of Hong Kong.

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

# A. Details on the General Points Environment

In this section, we demonstrate the design details for `GeneralPoints` mentioned in Section 4.1. We first present the data used for this environment (Appendix A.1). Then, we show examples of the environment's transition dynamics (Appendix A.2), followed by a description of key arguments and reward design specification (Appendix A.3).

## A.1. Data

`GeneralPoints` card quadruples are sampled from a deck of 52 standard poker cards. Each sampled quadruple is guaranteed to have at least one solution equals the target point, i.e. 24. We ensure this by using an expert solver during the sampling process.

## A.2. Detailed Examples on the Transition Dynamics

As shown in Figure 11 and Figure 12, we treat the system prompt as $v_0^{\text{in}}$ and then subsequently appending the future outputs $v_{1:t}^{\text{out}}$ and verifier info $v_{1:t}^{\text{ver}}$ into the prompt for getting the $t+1$ output. Figure 11 provides an example with the visual inputs, while Figure 12 shows the language only case.

## A.3. Additional Eetails on the Environmental Design

**Arguments.** The `GeneralPoints` environment supports the following configurable arguments:

- Target point: Any positive integer
- Face cards rule: Two options
  - 'J', 'Q', and 'K' all count as '10'
  - 'J', 'Q', and 'K' count as '11', '12', and '13' respectively
- Card sampling: Two options
  - Sample 4 cards without replacement from a deck of 52 poker cards
  - Sample at least one card from 'J', 'Q', and 'K'
- Card color: Three options
  - Black suits only: ♣, ♠.
  - Red suits only: ♥, ♦.
  - All suits: ♠, ♥, ♣, ♦.

For all experiments, we fix the target point at 24. In Figure 5, training and in-domain evaluation use the rule where face cards count as '10'. For out-of-domain evaluation, we use the alternative face cards rule and require at least one face card, forcing calculations with numbers above 10 that are not encountered during training. For visual distribution shift experiments (Section 5.2), we train the model on black suits ♠, ♣ and evaluate out-of-domain performance on red suits ♥, ♦.

**Reward design.** An episode terminates when either a correct equation is generated or the maximum verification step of 5 is reached. The reward function is as follows:

- $r = 5$: For generating a legal equation that equals the target point

- $r = -1$: For legal equations using each card once but not equaling the target point

- $r = -1$: For exceeding maximum verification step

- $r = -2$: For legal equations containing numbers not among the given choices

- $r = -3$: For all other illegal equations

In the vision-language variant (`GeneralPoints-VL`), an additional penalty of $r = -1.5$ is applied when the agent fails to correctly recognize the given cards.

# B. Details on the V-IRL Environment

Similar to Appendix A, we present the design details for `V-IRL` discussed in Section 4.2. First, we introduce the database used for this environment (Appendix B.1) and demonstrate transition examples (Appendix B.2). We then describe the environment by explaining its fundamental component—*route*. Finally, we outline our modifications and reward design choices made to adapt the original `V-IRL` for reinforcement learning training (Appendix B.3).

## B.1. Data

Leveraging the data collection pipeline of Yang et al. (2024a), we construct a training database with 1000 unique routes from New York City. We evaluate all rule-variant experiments and visual in-distribution experiments using randomly sampled routes from this database. For visual out-of-distribution experiments, we directly adopt the VLN mini benchmark from Yang et al. (2024a). This benchmark consists of 18 distinct routes across nine cities: Milan, New Delhi, Buenos Aires, London, Hong Kong, New York,[4] Melbourne, Lagos, and San Francisco, with two routes per city.

---

[4]These NYC routes in the VLN mini benchmark do not overlap with our training data.

**System Prompt ($v_0^{\text{in}}$)**

[Task Description]
You are an expert 24 points card game player. You are observing these four cards in the image. Note that 'J', 'Q', and 'K' count as '10', and each card must be used once. Your goal is to output a formula that evaluates to 24 using numbers from the cards and operators such as '+', '-', '*', '/', '(', ')', and '='.

[Output]
Your response should be a valid json file in the following format:
{
"cards": [x, y, z, w], where 'J', 'Q', and 'K' count as '10',
"number": [a, b, c, d], where a, b, c, and d are the numbers on the cards,
"formula": "an equation that equals 24",
}

---

**Appending model and verifier outputs to obtain $v_t^{\text{in}}$**

$v_t^{\text{in}} = [v_0^{\text{out}}, v_0^{\text{ver}}, v_1^{\text{out}}, v_1^{\text{ver}}, \ldots, v_{t-1}^{\text{out}}, v_{t-1}^{\text{ver}}]$ $\quad\quad\quad\quad\quad \triangleright\ v_t^{\text{in}} = \text{concat}\left(v_0^{\text{in}}, [v_k^{\text{out}}, v_k^{\text{ver}}]_{k=0}^{t-1}\right)$

---

**Model output ($v_t^{\text{out}}$)**

{
"cards": ['A', '3', 'K', '6'],
"number": [1, 3, 13, 6],
"formula": "(1+6)*3+13=24",
}

---

**Verifier Output ($v_t^{\text{ver}}$)**

You failed this trial because your formula is incorrect. $\quad\quad\quad\quad \triangleright\ v_{t+1}^{\text{in}} = \text{concat}(v_t^{\text{in}}, v_t^{\text{out}}, v_t^{\text{ver}})$

Figure 11: **An example of our prompt update** for constructing $v_{t+1}^{\text{in}}$ using $v_t^{\text{in}}$, $v_t^{\text{out}}$ and $v_t^{\text{ver}}$. This example provides an optional vision input for VLMs, adding a visual recognition challenge. The brown parts marks the task and related information, and the purple parts denote the state ($s_t$) specific info. The blue and red describe the output from the model and verifier, respectively.

## B.2. Detailed Examples on the Transition Dynamics

We provide detailed transition examples of the V-IRL environment in Figure 13 (vision and language) and Figure 14 (pure language).

## B.3. Additional Details on the Environmental Design

**Concept of *route*.** The route serves as the fundamental navigation object in the V-IRL environment. As illustrated in Figure 4, each route corresponds to a real-world path with associated language instructions and visual signals. Using Figure 4 as an example, a route comprises:

- Destination: Shuka

- Starting point: Start

- Turning points: The Dutch, Lola Taverna

- Straight road: Roads connecting turning points, starting point, and destination

- Street views: 360-degree panoramic views at each movable point

- Oracle information: Expert observation data for each movable point

- Expert trajectory

- Instruction

Although the instructions in Figures 4, 13 and 14 are presented in different formats, they convey equivalent information, with Figure 4 using natural language.

**Simplification and arguments.** We simplify the original V-IRL design from Yang et al. (2024a) to better accommodate RL training. The modifications include eliminating the 2-stage navigation pipeline that required a separate visual detector for street view processing, and removing online queries to reduce training time and cost. Our V-IRL environment contains 2 additional configuration arguments compared with the original design:

- Action space: two options

  - Absolute direction:
    "turn_direction(x)" where x∈{'north', 'northeast',

**System Prompt** ($v_0^{\text{in}}$)

[Task Description]
You are an expert 24 points card game player. You are observing these four cards in the image. Note that 'J', 'Q', and 'K' count as '11', '12', and '13' respectively, and each card must be used once. Your goal is to output a formula that evaluates to 24 using numbers from the cards and operators such as '+', '-', '*', '/', '(', ')', and '='.

[Input]
Cards: ['A', '3', 'K', '6']

[Output]
Your response should be a valid json file in the following format:
{
"cards": [x, y, z, w], where 'J', 'Q', and 'K' count as '10',
"number": [a, b, c, d], where a, b, c, and d are the numbers on the cards,
"formula": "an equation that equals 24",
}

─────────────────────────────────────────────────────────────────────

**Appending model and verifier outputs to obtain $v_t^{\text{in}}$**

$v_t^{\text{in}} = [v_0^{\text{out}}, v_0^{\text{ver}}, v_1^{\text{out}}, v_1^{\text{ver}}, \ldots, v_{t-1}^{\text{out}}, v_{t-1}^{\text{ver}}]$         $\triangleright\ v_t^{\text{in}} = \text{concat}\left(v_0^{\text{in}}, [v_k^{\text{out}}, v_k^{\text{ver}}]_{k=0}^{t-1}\right)$

**Model output** ($v_t^{\text{out}}$)

{
"cards": ['A', '3', 'K', '6'],
"number": [1, 3, 13, 6],
"formula": "(1+6)*3+13=24",
}

─────────────────────────────────────────────────────────────────────

**Verifier Output** ($v_t^{\text{ver}}$)

You failed this trial because your formula is incorrect.         $\triangleright\ v_{t+1}^{\text{in}} = \text{concat}(v_t^{\text{in}}, v_t^{\text{out}}, v_t^{\text{ver}})$

Figure 12: **An example of our prompt update** for constructing $v_{t+1}^{\text{in}}$ using $v_t^{\text{in}}$, $v_t^{\text{out}}$ and $v_t^{\text{ver}}$. This example provides an optional vision input for VLMs, adding a visual recognition challenge. The brown parts marks the task and related information, and the purple parts denote the state ($s_t$) specific info. The blue and red describe the output from the model and verifier, respectively.

'east', 'southeast', 'south', 'southwest', 'west', 'northwest'}, "forward()", "stop()"

- Relative direction:
  "turn_direction(x)" where x∈{'left', 'right', 'slightly left', 'slightly right'}, "forward()", "stop()"

- Maximum straight road length: any positive integer

The action space argument accommodates the rule variants described in Section 4. For experiments shown in Figure 5, we use absolute direction action space during training and in-domain evaluation, while using the alternative rule for out-of-domain evaluation. We implement a maximum straight road length to limit the number of movable coordinates between turning points, preventing sequences of repetitive "forward()" actions. We conduct visual distribution shift experiments (Section 5.2) via training the model on New York City regions and evaluating the out-of-domain performance on the worldwide navigation routes from the benchmark released by Yang et al. (2024a).

**Reward design.** An episode terminates when either the navigation agent stops at the destination or the maximum verification step of 2 is reached. The reward function is as follows:

- $r = 1$: For generating a correct action at the current coordinate

- $r = -1$: For generating wrong action at the current coordinate

- $r = -1$: For exceeding maximum verification step

- $r = -1.5$: For failed detection of landmarks

**System Prompt ($v_0^{\text{in}}$)**

[Task Description]
You are an expert in navigation. You will receive a sequence of instructions to follow while observing your surrounding street views. You are also provided with your observation and action history in text. your goal is to take the action based on the current observation and instruction.

[Instruction]
1. First, turn left to face east.
2. Move forward until you reach the next intersection where Hotel 32One is on your right behind.
3. Turn right to face north.
4. Move forward until you reach the next intersection where Dragon Gate Chinatown SF is on your right front.
5. Turn left to face east.
6. Move forward until the destination Café de la Presse is on your right.

[Current observation]
You observe a 2x2 grid of street view images with the following headings:
[front, right
 back, left]
You need to identify if any of the landmarks in the instruction are visible in the street view grid.

[Action space]
- "forward()": indicates moving forward for 1 step;
- "turn_direction(x)": indicates turn direction to the target heading, where x∈['north', 'northeast', 'east', 'southeast', 'south', 'southwest', 'west', 'northwest'];
- "stop()": indicates the navigation is finished;

[Observations and actions sequence]
O_1: No landmarks nearby;
A_1: turn_direction(east)
O_2: No landmarks nearby;
A_2: forward()
O_3: No landmarks nearby;
A_3: forward()
O_4: You observe an image of 4 views; You observe an intersection
A_4:

[Output]
Your response should be a valid json file in the following format:
{
"current observation": latest observation from the street view grid,
"current instruction": analyze the full instruction and identify the sentence to be executed,
"action": the action to be taken chosen from the action space,
}

-------------------------------------------------------------------------------------------------------
**Appending model and verifier outputs to obtain $v_t^{\text{in}}$**

$v_t^{\text{in}} = [v_0^{\text{out}}, v_0^{\text{ver}}, v_1^{\text{out}}, v_1^{\text{ver}}, \ldots, v_{t-1}^{\text{out}}, v_{t-1}^{\text{ver}}]$        $\triangleright\ v_t^{\text{in}} = \text{concat}\left(v_0^{\text{in}}, [v_k^{\text{out}}, v_k^{\text{ver}}]_{k=0}^{t-1}\right)$

-------------------------------------------------------------------------------------------------------
**Model output ($v_t^{\text{out}}$)**

{
"current observation": "Hotel 32One is on my right behind; I observe an intersection",
"current instruction": "Turn right to face north.",
"action": "turn_direction(north)",
}

-------------------------------------------------------------------------------------------------------
**Verifier Output ($v_t^{\text{ver}}$)**

Correct solution.        $\triangleright$ Move to the next scene upon correct action

Figure 13: **An example of our prompt update** for constructing $v_{t+1}^{\text{in}}$ using $v_t^{\text{in}}$, $v_t^{\text{out}}$ and $v_t^{\text{ver}}$. This example provides an optional vision input for VLMs, adding a visual recognition challenge. The brown parts marks the task and related information, and the purple parts denote the state ($s_t$) specific info. The blue and red describe the output from the model and verifier, respectively.

**System Prompt ($v_0^{\text{in}}$)**

[Task Description]
You are an expert in navgation. You will receive a sequence of instructions to follow. You are also provided with your observation and action histroy in text. Your goal is to first analyze the instruction and identify the next sentence to be executed. Then, you need to provide the action to be taken based on the current observation and instruction.

[Instruction]
1. First, turn left to face east.
2. Move forward until you reach the next intersection where Hotel 32One is on your right behind.
3. Turn right to face north.
4. Move forward until you reach the next intersection where Dragon Gate Chinatown SF is on your right front.
5. Turn left to face east.
6. Move forward until the destination Café de la Presse is on your right.

[Action space]
- "forward()": indicates moving forward for 1 step;
- "turn_direction(x)": indicates turn direction to the target heading, where x∈['north', 'northeast', 'east', 'southeast', 'south', 'southwest', 'west', 'northwest'];
- "stop()": indicates the navigation is finished;

[Observations and actions sequence]
O_1: No landmarks nearby;
A_1: turn_direction(east)
O_2: No landmarks nearby;
A_2: forward()
O_3: No landmarks nearby;
A_3: forward()
O_4: Hotel 32One is on your right behind; You observe an intersection
A_4:

[Output]
Your response should be a valid json file in the following format:
{
"current observation": latest observation from the street view grid,
"current instruction": analyze the full instruction and identify the sentence to be executed,
"action": the action to be taken chosen from the action space,
}

------------------------------------------------------------------------------------------------------------------------

**Appending model and verifier outputs to obtain $v_t^{\text{in}}$**

$v_t^{\text{in}} = [v_0^{\text{out}}, v_0^{\text{ver}}, v_1^{\text{out}}, v_1^{\text{ver}}, \ldots, v_{t-1}^{\text{out}}, v_{t-1}^{\text{ver}}]$    ▷ $v_t^{\text{in}} = \text{concat}\left(v_0^{\text{in}}, [v_k^{\text{out}}, v_k^{\text{ver}}]_{k=0}^{t-1}\right)$

**Model output ($v_t^{\text{out}}$)**

{
"current observation": "Hotel 32One is on my right behind; I observe an intersection",
"current instruction": "Turn right to face north.",
"action": "turn_direction(north)",
}

------------------------------------------------------------------------------------------------------------------------

**Verifier Output ($v_t^{\text{ver}}$)**

Correct solution.    ▷ Move to the next scene upon correct action

Figure 14: **An example of our prompt update** for constructing $v_{t+1}^{\text{in}}$ using $v_t^{\text{in}}$, $v_t^{\text{out}}$ and $v_t^{\text{ver}}$. The brown parts marks the task and related information, and the purple parts denote the state ($s_t$) specific info. The brown parts marks the task and related information, and the purple parts denote the state ($s_t$) specific info. The blue and red describe the output from the model and verifier, respectively.

# C. Experimental Setup

This section details the experimental setup used in Section 5. We first describe our data collection setup for supervised fine-tuning (Appendix C.1). Then, we present the training pipeline (Appendix C.2). Finally, we describe our evaluation metrics and the statistical tools used for generating plots (Appendix C.3).

## C.1. Data

**SFT data collection.** As illustrated in Figures 11 to 14, `GeneralPoints` and `V-IRL` environments naturally align with prompt-response dialogue structures. We create training samples by pairing each system prompt with its corresponding expert response. All SFT experiments in the main body use optimal single-turn prompt-response pairs, without any verification or revision steps.

**SFT on sub-optimal trajectories** To examine how more diverse SFT data affects the out-of-distribution performance of SFT, we conduct an ablation study on `GP-L` using sub-optimal trajectories as training data. Unlike expert prompt-response pairs, these sub-optimal trajectories include errors and verification messages in their prompts. This format aligns with evaluation scenarios where multiple verification iterations are allowed, similar to the data being used for the downstream RL training. In Figure 15, we observe that SFT still merely memorizes the training data with degraded out-of-distribution performance. This evidence suggests that memorization occurs due to the fundamental nature of SFT training rather than the SFT data.

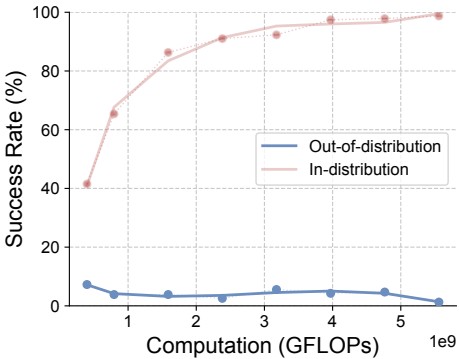

Figure 15: **SFT experiments on `GP-L` with suboptimal trajectories.** Similar to results in Figure 5, SFT overfits the training data even we increase the trajectory diversity.

## C.2. Training Pipeline

As illustrated in Section 5, we follow the training pipeline by RL4VLM (Zhai et al., 2024a), where we first initial-

ize the model with SFT, then separately scale up the compute for SFT and RL (Schulman et al., 2017), starting from this initialized model. For all experiments of SFT and RL in the main body, we tune all components using a shared learning rate per experiment. All training experiments are conducted on an 8 H800 machine (80GB).

## C.3. Evaluation Metric

**Per-step accuracy.** We report the per-step accuracy for `V-IRL-VL` task in Figures 5 and 6. An individual step is considered correct when the model's chosen action matches the expert trajectory at that position. Note that intermediate verification steps are counted as independent samples here.

**Success rate.** We report the success rate (%) of `GP-L`, `GP-VL`, `V-IRL-L` and `V-IRL-VL` in Figures 5 and 6. In the `GeneralPoints` task, success is defined as succeeding at least once during the inference time verification. In the `V-IRL` task, a sample is recorded as success when the model takes correct action at each movable point on the route.

**Computation estimation.** We estimate the FLOPs for training $X$ following the similar manner of (Snell et al., 2024; Hoffmann et al., 2023), where $X_{train} = 6ND_{train}$ and $X_{inference} = 2ND_{inference}$. Here, $N$ represents the model parameters and $D_{train}$ represents the number of tokens during training. Suppose our SFT and RL experients starts from a checkpoint trained on $D_{init}$ tokens, we can estimate the training computation of SFT and RL via the following equations:

$$X_{SFT} = 6N(D_{init} + D_{SFT})$$
$$X_{RL} = 6N(D_{init} + D_{RL}) + 2ND_{buffer}$$

Note that the used on-policy RL algorithm PPO (Schulman et al., 2017) contains iterative stages of replay buffer collection and optimization, hence requiring additional inference computation. For simplicity, we approximate the term via:

$$D_{buffer} \approx \frac{E\bar{d}_i\bar{d}_o}{D_{RL}} \cdot D_{RL}$$
$$= \lambda D_{RL}$$

where $E \in \mathbb{N}$ denotes the number of auto-regressive generation processes, $\bar{d}_i, \bar{d}_o$ denote average input tokens and output tokens. We estimate the $\lambda$ for `GeneralPoints` and `V-IRL` as 6 and 5.1 respectively after calculation.

**Line smoothing and error bar.** All line plots in our paper adopt Savitzky–Golay filter with polynomial order 3 as smoothing function. We assume each evaluated data point

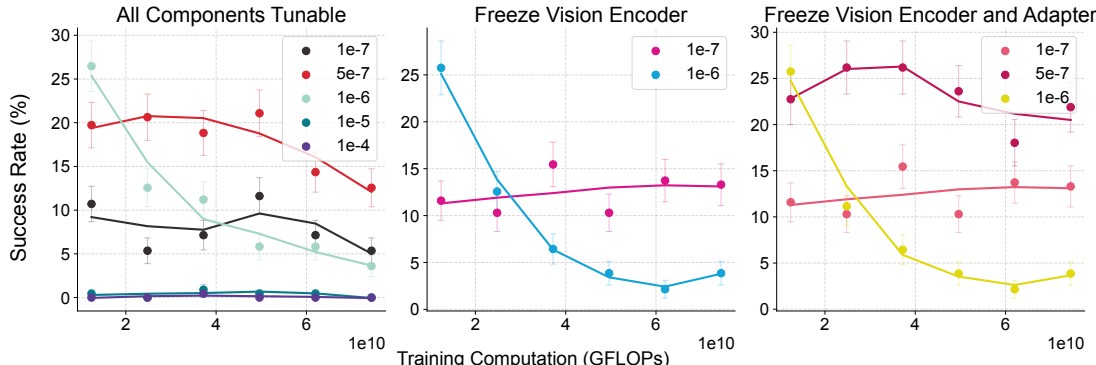

Figure 16: **Ablation studies on `GeneralPoints-VL` SFT.** We ablate the learning rate and report the in-distribution episode success rate (%) of all experiments. None of the experiments shows an increasing trend beyond 30% success rate.

follows a binomial distribution and approximate the standard error using $\sqrt{\frac{P(1-P)}{N}}$, where $P$ is the demical success rate and $N$ is the number of samples.

## D. Additional Experimental Results

In this section, we provide additional experimental results that are not covered in the main body.

### D.1. Ablation Studies on `GP-VL`

As mentioned in Section 6, we observe an abnormal phenomenon that SFT fails to achieve comparable in-distribution performance with RL (see Figure 5 subplot row 1 column 3). To further explore this, we conduct ablation studies over different hyperparameter choices.

**SFT.** We ablate the hyperparameter choices under the same task setting of `GP-VL` in Section 5.1. For experiments fine-tuning all parameters, we search learning rates from $\{1 \times 10^{-4}, 1 \times 10^{-4}, 1 \times 10^{-5}, 1 \times 10^{-6}, 5 \times 10^{-7}, 1 \times 10^{-7}\}$. Freezing the vision encoder, we search learning rates $\{1 \times 10^{-6}, 1 \times 10^{-7}\}$. Freezing vision encoder and adapter, we search learning rates $\{1 \times 10^{-6}, 5 \times 10^{-7}, 1 \times 10^{-7}\}$. We provide the in-distribution success rate curve in Figure 16.

**RL.** Finding suitable hyperparameters for RL experiments requires minimal effort. We conduct a search over learning rates $2 \times 10^{-6}, 1 \times 10^{-6}$, with the in-distribution success rate curves shown in Figure 17. All parameters are tunable in our RL experiments.

### D.2. More results on `V-IRL-VL`

Echoing per-step accuracy results in Figure 5, we report the overall success rate of `V-IRL-VL` in Figure 18. Due to the task's complexity, both training methods achieve overall success rates no higher than 1%. For `V-IRL`, the overall

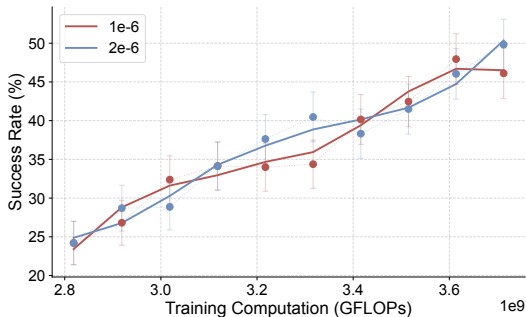

Figure 17: **Ablation studies on `GeneralPoints-VL` RL.** Echoing Figure 16, we ablate the learning rate and rreport the in-distribution episode success rate (%) of the two experiments. All components are tunable here.

success rate is a significantly more demanding metric since it aggregates per-step errors. For example, a random policy achieving 10% per-step accuracy would achieve achieve only approximately $10^{-8}\%$ success rate on enough routes averaging 10 steps in length.

### D.3. Failure Cases

In this section, we present 2 failure cases in our experiments as mentioned in Sections 5.4 and 6.

**Without SFT, RL fails.** In Figure 9, we present the training dynamics of failed RL experiments without SFT initialization. We additionally provide output examples of these experiments in Figure 20, where the model tends to generate unstructured response and fail.

**RL cannot save overfitted checkpoints.** As shown in Figure 19, RL cannot recover the out-of-distribution performance when initialized from a extremely overfitted checkpoint that has an initial per-step accuracy of less than 1%. We additionally provide an output example in Figure 19, where the model fails to adjust to the new rule.

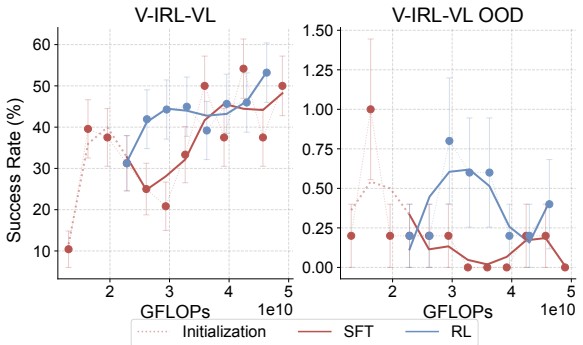

Figure 18: **Overall success rate (%) - GFLOPs for V-IRL-VL under rule variants.** Due to the nature of the task requiring aggregating a trajectory of correct actions, neither training method achieves reasonable out-of-distribution performance.

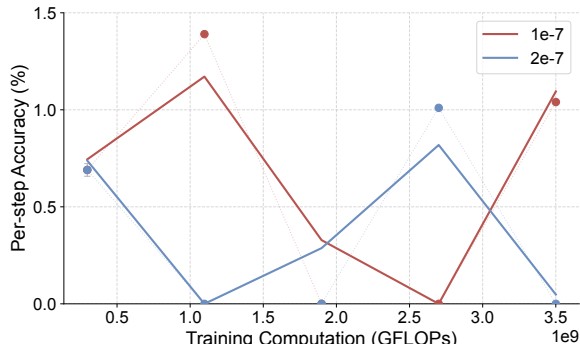

Figure 19: **Out-of-distribution per-step accuracy (%) - GFLOPs for V-IRL-VL under rule variants with overfitted initial checkpoint.** Evaluation metric details can be found in Appendix C.3.

---

**System Prompt ($v_0^{\text{in}}$)**

[Task Description]
You are an expert 24 points card game player. You are observing these four cards in the image. Note that 'J', 'Q', and 'K' count as '11', '12', and '13' respectively, and each card must be used once. Your goal is to output a formula that evaluates to 24 using numbers from the cards and operators such as '+', '-', '*', '/', '(', ')', and '='.

[Input]
Cards: ['2', 'J', '3', 'K']

[Output]
Your response should be a valid json file in the following format:
{
"cards": [x, y, z, w], where 'J', 'Q', and 'K' count as '10',
"number": [a, b, c, d], where a, b, c, and d are the numbers on the cards,
"formula": "an equation that equals 24",
}

---
**Appending model and verifier outputs to obtain $v_t^{\text{in}}$**

$v_t^{\text{in}} = [v_0^{\text{out}}, v_0^{\text{ver}}, v_1^{\text{out}}, v_1^{\text{ver}}, \ldots, v_{t-1}^{\text{out}}, v_{t-1}^{\text{ver}}]$  $\triangleright v_t^{\text{in}} = \text{concat}\left(v_0^{\text{in}}, [v_k^{\text{out}}, v_k^{\text{ver}}]_{k=0}^{t-1}\right)$

---
**Failed model output ($v_t^{\text{out}}$)**

To solve this problem, we can use a brute force approach by generating all possible combinations of the cards and checking if any of them can be used to form an equation that equals 24.
Here is a Python solution for this problem:
```python
def calculate_formula(cards):
# Define the mapping of cards to their corresponding numbers
```
. . .

---
**Verifier Output ($v_t^{\text{ver}}$)**

You failed this trial because your formula is incorrect.  $\triangleright v_{t+1}^{\text{in}} = \text{concat}(v_t^{\text{in}}, v_t^{\text{out}}, v_t^{\text{ver}})$

---

Figure 20: **Example model outputs without SFT initialization.** We record model responses using prompts similar to the one shown in Figure 11. The results demonstrate that Llama-3.2-Vision-11B fails to follow instructions properly. We omit the long response which tries to solve the puzzle via code but fails to finish within finite context length.

**System Prompt ($v_0^{\text{in}}$)**

[Task Description]
You are an expert in navigation. You will receive a sequence of instructions to follow while observing your surrounding street views. You are also provided with your observation and action history in text. Your goal is to first analyze the instruction and identify the next sentence to be executed. Then, you need to provide the action to be taken based on the current observation and instruction.

[Instruction]
1. First, turn right to face northwest.
2. Move forward until you reach next intersection where Korean War Memorial is on your left.
3. Turn left to face southwest.
4. Move forward until you reach next intersection where Korean War Memorial is on your left behind.
5. Turn right to face north.
6. Move forward until you reach next intersection.
7. Turn left to face east.
8. Move forward until you reach next intersection.
9. Turn left to face north.
10. Move forward until you reach next intersection.
11. Turn right to face east.
12. Move forward until you reach next intersection.
13. Turn left to face northeast.
14. Move forward until you reach next intersection where 9/11 Memorial & Museum is on your left.
15. Turn right to face northwest.
16. Move forward until you reach destination where The destination 9/11 Memorial & Museum is on your right front.

[Action space]
"forward()": indicates moving forward one step
"turn_direction(x)": indicates adjust the ego agent direction towards x direction. x could be any following ['left', 'right', 'slightly left', 'slightly right']
"stop()": indicates the navigation is finished.

[Observations and actions sequence]
O_1: You observe an image of 4 views;
A_1:

[Output]
{
"current observation": latest observation from the streetview grid,
"current instruction": analyze the full instruction and identify the sentence to be executed,
"action": the action to be taken chosen from the action space,
}

-------------------------------------------------------------------------------------------------------------------

**Appending model and verifier outputs to obtain $v_t^{\text{in}}$**

$v_t^{\text{in}} = [v_0^{\text{out}}, v_0^{\text{ver}}, v_1^{\text{out}}, v_1^{\text{ver}}, \ldots, v_{t-1}^{\text{out}}, v_{t-1}^{\text{ver}}]$        $\triangleright\ v_t^{\text{in}} = \text{concat}\left(v_0^{\text{in}}, [v_k^{\text{out}}, v_k^{\text{ver}}]_{k=0}^{t-1}\right)$

-------------------------------------------------------------------------------------------------------------------

**Failed model output ($v_t^{\text{out}}$)**

{
"current observation": "No landmarks nearby",
"current instruction": "Turn right to face northwest.",
"action": "turn_direction(northwest)",
}

-------------------------------------------------------------------------------------------------------------------

**Verifier Output ($v_t^{\text{ver}}$)**

Incorrect action.        $\triangleright$ Fail this sample.

Figure 21: **Failed example of** `V-IRL` **transition due to overfitting.** This phenomenon happens more frequently during scaling up supervised fine-tuning.

