# OpenReview forum: "SFT Memorizes, RL Generalizes: A Comparative Study of Foundation Model Post-training"
_ICML.cc/2025/Conference — ICML 2025 poster_

### Official Review · Reviewer_KPrc · 2025-03-12

**Overall Recommendation:** 2

**Summary:**

This paper conducts a comparative study of SFT and RL for post-training on GeneralPoints, an arithmetic reasoning card game, and also considers V-IRL, a real-world navigation environment. Experimental results show RL leads to model generalizing better in OOD cases while models trained with SFT hardly generalize.  The authors also discuss the role of SFT by concluding that SFT is still necessary to
stabilize the model’s output format, enabling RL to achieve its performance gains.

**Claims And Evidence:**

1. SFT memorizes, RL generalizes. This is the main claim of this paper and is a very strong claim. I think it deserves more rigorous evidence under more controllable setups. The author concludes this by experimenting with GeneralPoints and V-IRL. They train models to learn the tasks with SFT and RL respectively and evaluate the performance in synthetic OOD settings. The synthetic nature ensures the validity of OOD scenarios. Nonetheless, the learning setups are questionable as the quality of SFT data is well controlled. It is unclear whether the gaps arise from RL learning more generalizable CoTs/trajectories.

2. Scaling RL improves visual recognition accuracy in VLM training. The authors extend the OOD generalization benefits in visual tasks to study the underlying reasons and find that performance improvement correlates with visual recognition accuracy.

2. SFT is still necessary to stabilize the model’s output format. The author supports this by showing RL without SFT initialization fails.

**Essential References Not Discussed:**

Not found.

**Experimental Designs Or Analyses:**

See discussions about the claims.

**Methods And Evaluation Criteria:**

See discussions about the claims.

**Other Comments Or Suggestions:**

It would be better to have better control of the SFT data and conduct a more thorough comparison of different SFT trajectories.

For example, the authors can SFT by distillation from the RL models to ensure the distribution of CoTs is consistent.

**Other Strengths And Weaknesses:**

All experiments in this paper are conducted with fixed and ~7B-scaled models. It is unclear how larger pretrained models affect the generalization of SFT and RL.

**Questions For Authors:**

Have you experimented with larger models and are there any generalization benefits brought by better pretraining?

**Relation To Broader Scientific Literature:**

This paper correlates to the study about the generalizability and training efficiency of data from different sources. For instance, (1) fine-tuning with on-policy synthetic data (RL) enhances learning efficiencies in math reasoning [1].

[1] RL on Incorrect Synthetic Data Scales the Efficiency of LLM Math Reasoning by Eight-Fold

**Theoretical Claims:**

N/A

---

> ### Author Rebuttal · Authors · 2025-04-01
>
> ## General Response
> Dear reviewer KPrc,
>
> We sincerely thank you for your valuable feedback. We especially appreciate your advice on making the claim more rigorous. To best of our effort in the rebuttal period, we conduct the following experiments to strengthen our evidence:
> - Experiments on Qwen-2.5-VL-3B. See results in Figure 22 of [rebuttal material](https://drive.google.com/file/d/1WheCe-fkbX7jLKn2hsO7E701nGEPkuhJ).
> - Experiments on Llama-3.2 when fine-tuned by distillation from the RL models. See results in Figure 24 of [rebuttal material](https://drive.google.com/file/d/1WheCe-fkbX7jLKn2hsO7E701nGEPkuhJ).
>
> We provide the following feedback for your specific concerns.
>
> > Q1. Concerns on experiments with different model sizes.
>
> Thank you for your suggestion. We conduct experiments on [Qwen-2.5-VL-3B](https://huggingface.co/Qwen/Qwen2.5-VL-3B-Instruct), an up-to-date MLLM with smaller model size. We observe that **SFT memorizes, RL generalizes still holds in this case.** Specifically, RL achieves an increase of +3.45% on OOD while SFT causes a drop of 8.48%. Detailed performance curve can be found in Figure 22 of [rebuttal material](https://drive.google.com/file/d/1WheCe-fkbX7jLKn2hsO7E701nGEPkuhJ). Experimental settings are the same as our original paper except we train fewer steps due to time constraint.
> We are also interested in scaling up our method to larger models. But due to resource constraints, we are unable to conduct experiments on larger models (>=32B) during the rebuttal period.
>
> > Q2. Concerns on diverse trajectories
>
> Good point and we add a group of experiments on distillation of the RL model. As is illustrated in Figure 24 of [rebuttal material](https://drive.google.com/file/d/1WheCe-fkbX7jLKn2hsO7E701nGEPkuhJ), **we observe faster in-distribution performance increase with less out-of-distribution degradation compared to the original SFT experiments**. This evidence still aligns with our original finding but demonstrates the positive effect of diverse SFT data. We appreciate you for pointing this out and we believe that “generalization by comparing long CoT + RL (deepseek / o1) vs well curated data for SFT” could be another important question to study in the future.
>
> Once again, we would like to thank the reviewer again for insightful and careful suggestions, if you feel like our response and additional results further improve the quality of our work, please feel free to improve your rating, thank you very much in advance!.

---

> > ### Comment · Reviewer_KPrc · 2025-04-08
> >
> > Thank you so much for your responses and the supplemented experiments.
> >
> > Given the results of Fig. 24, I think it does reflect that what CoT (e.g., on-policy v.s. off-policy, detailed v.s. shallow) to learn is probably one of the decisive factors on generalization. Given the results, I still think the claim "SFT memorizes, RL generalizes" is overly sensational and requires more rigorous and in-depth analysis to present the true underlying mechanism. Therefore, I lean toward keeping my rating.

---

### Official Review · Reviewer_NgMz · 2025-03-13

**Overall Recommendation:** 2

**Summary:**

This paper studies the generalization of RL and SFT.
It uses two visual-language reasoning tasks, and shows that RL has better generalization and SFT mainly memorizes the training samples and struggles with the OOD samples.
Further analysis shows that RL can also improve the model's underlying visual recognition capabilities.
Despite RL's superior generalization, SFT is still helpful in stabilizing the output format.

**Claims And Evidence:**

The paper studies an important problem of what are the different roles of RL and SFT in generalization of LLM post-training.
The claim is that RL generalizes and SFT memorizes and the provided evidences are experiments on GeneralPoint Game and V-IRL task.
Their results show that the learning of RL can be better generalized to their chosen OOD tests.

I understand that the authors want to have a controlled experiment, but the experimental setting weakens the reliability of your conclusion.
- As you mentioned, your RL is based on the SFT trained model. Does this mean that your base model for RL and SFT are different? (RL starts from SFT-trained LLaMA and SFT starts from LLaMa). If it is so, I think your experimental results cannot support a comparison between RL and SFT.
- Your setting couples too many factors, making the experimental results hard to understand. For example, why do you consider a sequential revision instead of directly outputting the answer in the GeneralPoint Game. This introduces an additional factor of whether the model can correct its own results. A cleaner and simpler setting would help much. Why don't you experiment on more well-studied scenarios like math or other reasoning tasks.
- Have you tested other sizes of models or other training steps of the training periods? It seems the SFT training gains the improvement on in-domain tests in the early stages of SFT training, is your observation related to overfitting of SFT training?

**Essential References Not Discussed:**

NA

**Experimental Designs Or Analyses:**

In most of the experiments, I would say the performance of RL on OOD test sets does not drastically drop rather than "generalization".
For example, in GP, the OOD performance of RL is merely 12 to 17, and the in-domain acc is 50+. The improvements are marginal.

**Methods And Evaluation Criteria:**

To some extent.

**Other Comments Or Suggestions:**

NA

**Other Strengths And Weaknesses:**

NA

**Questions For Authors:**

- Previous research has shown that smaller model sizes tend to memorize, while larger models are capable of generalization [1]. Does the discovery in this paper apply specifically to a limited model size? For instance, would even smaller models, such as 0.5B or 2B parameters, exhibit memorization in both the SFT and RL periods? Additionally, do larger models, such as 72B or beyond, demonstrate generalization?

- Recent studies have found that when the diversity and quality of SFT data is sufficient, models can also learn generalized knowledge [2]. Does this paper potentially overclaim its findings, given that the cost of exploration during the RL phase is significantly higher than in the SFT phase?

[1] Generalization v.s. Memorization: Tracing Language Models’ Capabilities Back to Pretraining Data

[2] LIMO: Less is More for Reasoning

**Relation To Broader Scientific Literature:**

N

**Theoretical Claims:**

N/A

---

> ### Author Rebuttal · Authors · 2025-04-01
>
> ## General Response
> Dear reviewer NgMz,
>
> Thank you for your appreciation of our work, especially the importance of our studied problem. We also acknowledge your constructive feedback on improving the simplicity of our works. Here is our feedback:
>
> > Q1. Do RL and SFT start from different checkpoints?
>
> A quick clarification is that – while our RL starts from a SFTed LLama checkpoint (say `ckpt A`) for warmup, we still continue the future SFT from `ckpt A`. So we believe our experiments are still fair, as we scale up RL and SFT training flops from a SFTed LLama checkpoint.
>
> For more details, say in the top left picture in Figure 5, we started both RL & continue SFTed on the second leftmost point, which have been SFTed for 1.6e9 GFLOPs, and the dotted curves are the SFTed warmup flops. This consistent initialization ensures a fair comparison between the two approaches. The performance of these base models (`ckpt A`) is recorded as "init" in Figure 6, providing a clear baseline for measuring the relative improvements from each method.
>
> > Q2. Concerns about experimental complexity and task selection
>
> We appreciate your suggestion for a simpler setup.
>
> The sequential revision framework is chosen to align with multi-turn reinforcement learning framework. Our experiments also involve the case without sequential-revision demonstrated in Section 5.5, Figure 10, noted as VIter 1. Generalization is also observed here.
>
> The rule-based decision-making tasks allow us to conduct controlled experiments via switching different rules. Math problems, though well-studied, remain difficult to design out-of-distribution scenarios. We thank you for this insightful point and will continue to figure out rigorous ID/OOD tests for these tasks in future works.
>
> > Q3. Effect of model sizes & different training steps
>
> Thanks for the suggestion and we do more experiments:
>
> - **RL generalizes holds across different model sizes**: We conduct experiments on [Qwen-2.5-VL-3B](https://huggingface.co/Qwen/Qwen2.5-VL-3B-Instruct), where we find that **“SFT memorizes, RL generalizes” still holds for this model**. See results in Figure 22 in [rebuttal material](https://drive.google.com/file/d/1WheCe-fkbX7jLKn2hsO7E701nGEPkuhJ).
> - **RL generalizes holds across checkpoints of different initializations**: We provide results of two starting checkpoints, initialized by different amounts of SFT. We observe that **RL consistently increases OOD performance**. See results in Figure 23 in [rebuttal material](https://drive.google.com/file/d/1WheCe-fkbX7jLKn2hsO7E701nGEPkuhJ).
>
> > Q3.1. Is the evidence of SFT related to overfitting?
>
> Yes, memorization exactly equals to overfitting in our work. The early improvements on ID tests directly support the evidence, where the model rapidly memorizes and overfits the training data while forgetting out-of-domain knowledge.
>
> > Q4. Concerns about OOD improvements
>
> We appreciate your concern on the absolute OOD performance. The purpose of our study is to compare the behavior of SFT and RL instead of pursuing high performance increases. We believe “RL generalizes” holds as we observe decent percentages of ID increases transfer to OOD increases for RL with details provided in the table:
> | Task | GP-L | V-IRL-L | GP-VL | V-IRL-VL |
> |------|------|---------|-------|----------|
> | ID increase |+15.3%|+15.0%|+27.4|+3.29%|
> | OOD increase |+3.5%|+11.0%|+3.0%|+9.3%|
> | OOD improve / ID increase|22.9%|73.3%|10.9%|282.7%|
>
> **As opposed to the performance improvement in scaling up SFT, which results in performance decrease in OOD tasks.**
>
> More evident OOD performance growth happens when we scale up the revision iterations. In Figure 10, we observe that 55% of ID increases transfer to OOD when scaling up the number of iteration to 10 on GP-L task.
>
> > Q5. Regarding evidences in LIMO
>
> We thank the reviewer for mentioning a great concurrent work. In our humble opinion, we believe our results do not contradict with the LIMO, due to the different focus of our studies. Our focus is on “comparing generalization between SFT on ground truth data versus running vanilla end-to-end RL”, whereas LIMO [1] demonstrates that SFT on “well-curated data can also achieve remarkable performance on mathematical reasoning tasks”, similar to the recently released s1 paper [2]. Slightly extended to your insightful question, we believe that “generalization by comparing long CoT + RL (deepseek / o1) vs well curated data for SFT” could be another important question to study in the future.
>
> Once again we would like to thank the reviewer again for insightful and careful suggestions, if you feel like our response and additional results further improve the quality of our work, please feel free to improve your rating, thank you very much in advance!
>
> > References
>
> [1] Ye et al., 2025. LIMO: Less is More for Reasoning. arXiv preprint arXiv:2502.03387.
>
> [2] Muennighoff et al., 2025. s1: Simple test-time scaling. arXiv preprint arXiv:2501.19393.

---

> > ### Comment · Reviewer_NgMz · 2025-04-08
> >
> > Thank the authors for the detailed rebuttal. The response resolves some of my concerns about settings.
> >
> > Still, as I mentioned in my review about overfitting, training steps, and model size, I think better SFT setups with carefully curated data can mitigate the claimed result that SFT does not generalize. Also, the conclusion that SFT has a generalization issue is not exciting and can be found even in ML textbooks.
> >
> > For the above reasons, I will keep my score.

---

### Official Review · Reviewer_Dv4H · 2025-03-24

**Overall Recommendation:** 4

**Summary:**

This paper compares supervised fine-tuning (SFT) and reinforcement learning (RL) on both textual and visual reasoning tasks. The authors introduce GeneralPoints, an arithmetic reasoning card game, and V-IRL, a real-world navigation environment, to evaluate model generalization to unseen variants involving novel textual rules and visual domains.

The main results are:
- RL with outcome-based rewards, generalizes well to OOD scenarios in both two tasks, while SFT tends to memorize training data and struggles with OOD generalization.
- SFT serves as a useful initialization step for RL by stabilizing output formats and providing a solid foundation for effective RL training.
- RL training enhances the model’s underlying visual recognition abilities, contributing to better generalization in visual tasks.

Overall, the paper presents comprehensive experiments demonstrating that SFT is fundamental of stable RL training, and RL is effective at improving model's generalization in complex, multimodal reasoning environments.

**Claims And Evidence:**

The key claims are convincingly supported by clear experimental results and analyses. The authors are transparent about limitations, and most claims are well validated.

One potential caveat is that the claim “RL generalizes” holds under certain boundary conditions. The authors acknowledge that RL struggles to recover from overly-tuned SFT checkpoints (Section 6), indicating that RL’s generalization ability can be limited when applied to extremely underfit or overfit initial checkpoints.

**Essential References Not Discussed:**

The paper discusses the difference between memorization and generalization in large language models (LLMs), but does not sufficiently reference Wang et al., 2024 (“Generalization vs memorization: Tracing language models’ capabilities back to pretraining data”). This work systematically analyzes the relationship between model capabilities and pretraining data, providing direct theoretical support for the paper’s key conclusion that SFT tends to encourage memorization, while RL promotes generalization.

**Experimental Designs Or Analyses:**

While the experimental design is overall solid and well-motivated, I have several suggestions for improvement:
- The paper does not provide sensitivity analyses with respect to RL reward shaping, or PPO configurations.
- The paper heavily relies on a verifier-based reward signal but provides limited details on the verifier’s architecture. It is unclear whether the verifier is rule-based, learned, or manually engineered.

**Methods And Evaluation Criteria:**

The proposed methods and evaluation criteria are well-designed and appropriate for the problem at hand. The paper introduces two custom-designed tasks aimed at testing generalization:

- GeneralPoints, an arithmetic reasoning card game, includes both rule variations (to assess rule-based generalization) and visual variations (to assess visual generalization).

- V-IRL, a real-world navigation environment, involves complex spatial reasoning and visual recognition, with both action-space rule variations and visual distribution shifts.

The evaluation metrics—success rates for GeneralPoints and per-step accuracy for V-IRL—are well-defined and directly measure out-of-distribution generalization. In addition, the authors conduct scaling analyses (varying compute budgets and verifier iterations), making the evaluation more comprehensive and robust.

**Other Comments Or Suggestions:**

I have no additional comments or suggestions.

**Other Strengths And Weaknesses:**

Strengths

- Originality:
The paper presents a systematic comparative study between SFT and RL on both text-based and visual reasoning tasks.

- Significance:
The work addresses an important  question in foundation model research — how SFT and RL respectively contribute to generalization. The results offer valuable insights for designing post-training strategies in large-scale multimodal models.

- Clarity:
The paper is well-structured, with clear explanations, and informative figures and tables. The design decisions, and experimental setups are well-documented.

Weaknesses

- Limited scope of applicability:
While the authors demonstrate that RL benefits from SFT initialization, this finding is tested only on a single backbone (Llama3.2-Vision-11B). It remains unclear whether the same observations hold for other architectures and domains.

- Lack of deeper interpretability analysis:
Although the paper shows that RL improves visual recognition capability, it remains unclear whether this stems from changes in the visual encoder representations or purely from downstream policy optimization. More analysis at the representation level could strengthen this claim.

**Questions For Authors:**

- Your experiments are conducted on Llama3.2-Vision-11B. Have you observed similar SFT vs. RL generalization patterns on other model families or scales?
- You mention that RL cannot recover OOD performance when starting from an overfit SFT checkpoint. Could alternative RL reward shaping or curriculum design alleviate this problem?

**Relation To Broader Scientific Literature:**

The paper contributes to the broader scientific literature in several ways:

- Post-training techniques and generalization:

Prior works on SFT emphasize its role in improving model abilities.Similarly, reinforcement learning (RL) has been used for model alignment and human preference optimization. This paper extends the literature by systematically comparing SFT and RL in the context of generalization vs. memorization, filling a gap where most previous works only focus on one method or one modality.

- Scaling inference-time compute and verification:

This paper demonstrates that scaling verification iterations in RL training improves OOD generalization, providing further confirmation of inference-time compute scaling laws.

- Contribution to model interpretability:

The paper contributes to the broader discussion on model interpretability and reliability. Understanding the distinct roles of post-training techniques helps to know how generalization comes.

**Theoretical Claims:**

There are no theoretical claims or proofs to check in this submission. The paper’s contributions are empirical and experimental rather than theoretical.

---

> ### Author Rebuttal · Authors · 2025-04-01
>
> ## General Response
> Dear reviewer Dv4H,
>
> Thank you for your appreciation of our work. We are delighted to hear that you find our research original, significant, and clear. We provide the following feedback and additional experiments for your concerns:
> > Q1. Regarding Verifier design and PPO configuration
>
> We appreciate your carefulness on this point and we are with you that reward shaping and RL hyperparameters matter a lot in training.
>
> We provide the detailed reward & verifier design in Appendix A.3 and B.3 for GeneralPoints and V-IRL respectively. Specifically, we adopt rule-based rewards for these two tasks, where the model receives positive rewards if and only if it correctly solves the problem. For different failure cases, we intuitively set up different negative rewards as punishment. We implement a very simple verifier for all our settings, where the verifier function takes in string output responses and outputs different verification information based on rewards. You may kindly refer to Figure 2 for examples.
> We use a shared PPO configuration for all the experiments:
>
> | Parameter | Value |
> |------------|-------|
> | clip_param | 0.1 |
> | ppo_epoch | 4 |
> | value_loss_coef | 0.5 |
> | entropy_coef | 0.01 |
> | max_grad_norm | 0.01 |
> | gamma | 0.9 |
> | gae_lambda | 0.95 |
>
> We did not put much effort on engineering reward shapes and PPO configuration, as we directly adopted the PPO configuration from RL4VLM [1].
>
> > Q3. Are there experiments on other models?
>
> Yes, we conduct additional experiments on [Qwen-2.5-VL-3B](https://huggingface.co/Qwen/Qwen2.5-VL-3B-Instruct), an up-to-date SOTA MLLM from a different family. We adopt the same training setting as our original paper and plot the performance dynamic on Figure 22, [rebuttal material](https://drive.google.com/file/d/1WheCe-fkbX7jLKn2hsO7E701nGEPkuhJ). We observe that **“SFT memorizes, RL generalizes” still holds for this model**. Specifically, RL achieves an increase of +3.45% on OOD while SFT causes a drop of 8.48%. Due to resource constraints, we are unable to conduct experiments on larger models (>=32B) during the rebuttal period.
> > Q4. Deeper analysis?
>
> > Q4.1. on visual capabilities
>
> Insightful suggestion! We also find that there’s a lack of analysis on the visual encoder after being binded with the LLM, or furtherly trained by RL. We will explore this direction in our future work.
>
> > Q4.2. on alleviate the problem of overfitted checkpoints via reshape the rewards
>
> Interesting point! The answer may not be answered by simple yes or no but we have some thoughts on it.
>
> Remind that we design the reward purely based on outcomes. Considering an overfitted checkpoint with 100% ID acc and 0% OOD acc, the checkpoint will neither receive diverse reward signals nor be effectively updated during RL training as its response leads to uniform positive rewards. We think reshaping the values of reward functions does not help much in this case. On more general tasks or non-extreme cases, we believe that a careful design of reward functions will encourage exploration and benefit the generalization.
>
> > Q5. Regarding captions in figure 13 and 14 and literature to be cited
>
> Thank you so much for pointing this out! We will reorganize the figures and captions in our next revision. We read the paper by Wang et al., 2024 and also found it related to our research. We will cite it in our future revision as well.
>
> We would like to thank the reviewer again for insightful and careful suggestions, if you feel like our response and additional results further improve the quality of our work, please feel free to improve your rating, thank you very much in advance!
>
> > References
>
> [1] Zhai et al. "Fine-tuning large vision-language models as decision-making agents via reinforcement learning." Advances in Neural Information Processing Systems 37 (2024): 110935-110971.

---

> > ### Comment · Reviewer_Dv4H · 2025-04-08
> >
> > Thank you for the detailed response. The authors have addressed my concerns, and the additional clarifications are satisfactory. I will maintain my previous score and recommendation for acceptance.

---

### Official Review · Reviewer_rx4i · 2025-03-26

**Overall Recommendation:** 3

**Summary:**

This study compares the effects of supervised fine-tuning (SFT) and reinforcement learning (RL) on the post-training of foundation models, particularly in terms of generalization and memorization. It introduces two tasks -- GeneralPoints and V-IRL -- to evaluate how these techniques influence model performance in rule-based reasoning and visual domains. The results indicate that RL enhances generalization across both textual and visual tasks, while SFT tends to lead to memorization of the training data and struggles with out-of-distribution generalization. Further analysis reveals that RL improves the model’s visual recognition capabilities. Despite RL’s good generalization, SFT is found to be necessary for stabilizing the model’s output format. Additionally, the study shows that increasing the number of verification iterations during inference time improves RL’s generalization capability.

**Claims And Evidence:**

The claims in the submission are supported by experimental evidence. For instance, in the GeneralPoints task, RL achieved a +3.5% improvement in OOD performance, while SFT showed an -8.1% degradation. In the V-IRL task, RL improved OOD performance by +11.0%, whereas SFT experienced a -79.5% drop. These results suggest that RL enhances generalization across rule-based reasoning and visual tasks, while SFT may lead to memorization and struggle with out-of-distribution generalization. Additionally, the study found that RL improved visual recognition accuracy in VLMs, contributing to better overall performance. The authors also highlighted the necessity of SFT for stabilizing model outputs, which is crucial for effective RL training.

**Essential References Not Discussed:**

No

**Experimental Designs Or Analyses:**

The experimental designs and analyses are sound and valid. The tasks and environments are suited for evaluating generalization and memorization, and the metrics used provide a comprehensive assessment of model performance.

**Methods And Evaluation Criteria:**

The methods and evaluation criteria are appropriate for comparing SFT and RL in terms of generalization and memorization. The tasks and environments cover various reasoning and visual capabilities. The metrics used, such as success rate, per-step accuracy, and computational resources, offer a comprehensive assessment of model performance.

**Other Comments Or Suggestions:**

N/A

**Other Strengths And Weaknesses:**

Pros：
1. The study provides a comprehensive comparison of SFT and RL in terms of generalization and memorization, offering valuable insights into their respective strengths and limitations.

2. The introduction of GeneralPoints and V-IRL tasks is well-suited for evaluating rule-based reasoning and visual generalization, effectively testing models' ability to generalize beyond training data.

3. The study demonstrates state-of-the-art performance on the V-IRL mini benchmark, showcasing the effectiveness of the proposed RL approach.

Cons:
1. The study's focus on specific tasks (GeneralPoints and V-IRL) may limit the broader applicability of the findings, suggesting a need for more diverse experiments.

2. The paper includes many technical details, which might be challenging for readers unfamiliar with the field. Simplifying some sections and providing more intuitive explanations could enhance readability.

3. This is a purely experimental paper. Although it provides some experimental analysis, it lacks theoretical analysis, which is my main concern. I hope the authors can provide some theoretical analysis, e.g., why RL is superior for generalization than SFT ?

**Questions For Authors:**

See Strengths And Weaknesses

**Relation To Broader Scientific Literature:**

The contributions are related to prior research on post-training techniques, generalization, and visual capabilities. It investigates how SFT and RL affect model performance in rule-based and visual tasks, emphasizing the necessity of SFT for effective RL training. The study also explores the impact of scaling up inference-time computing on RL generalization, and shows that RL can enhance visual recognition in VLMs.

**Theoretical Claims:**

The submission does not include formal proofs or theoretical claims that require verification of mathematical correctness. The focus of the study is on empirical evaluation through experimental methods and analysis.

---

> ### Author Rebuttal · Authors · 2025-04-01
>
> ## General Response
> Dear reviewer rx4i,
> Thank you for your appreciation of our work. We are glad that you find our work comprehensive, well-designed, and recognize our SOTA results on V-IRL.
>
> Regarding your concerns, we provide the following feedback.
> > Q1. Suggestion on more diverse experiments for larger impact
>
> We agree with you that more diverse experiments will increase the impact of our work. For this purpose, we provide additional experiments in three folds:
> - Diversity on initial checkpoints: we provide additional experiments on [Qwen-2.5-VL-3B](https://huggingface.co/Qwen/Qwen2.5-VL-3B-Instruct), where **“SFT memorizes, RL generalizes” still holds for Qwen-2.5VL-3B.** Specifically, RL achieves an increase of +3.45% on OOD while SFT causes a drop of 8.48%. Experimental settings are the same as our original paper except we train fewer steps due to time constraint. See more detailed curves in Figure 22 in [rebuttal material](https://drive.google.com/file/d/1WheCe-fkbX7jLKn2hsO7E701nGEPkuhJ).
> - Diversity on training data: we conduct extra SFT experiments on diversified trajectories generated by both RLed checkpoints (Figure 24 in [rebuttal material](https://drive.google.com/file/d/1WheCe-fkbX7jLKn2hsO7E701nGEPkuhJ)) and synthetic approaches (Figure 15 in [original paper](https://openreview.net/pdf?id=dYur3yabMj)). In all three settings, we observe decreasing trends in OOD performance. Meanwhile, distillation on RLed checkpoints demonstrates faster in-distribution performance increase with less out-of-distribution degradation compared to the original SFT experiments.
>
> > Q2 Regarding readability and theory
>
> Thanks for your suggestions! We will make sure to update the manuscript, to provide a better introductory paragraph at the beginning of the experiment section for better readability. Regarding theory, we are actually actively investigating the theoretical explanations for such differences between RL and SFT, thanks for pointing out this insightful direction!
>
> Thank you again for your careful review and appreciation of our work. If you find our feedback alleviates your concerns, feel free to improve your rating accordingly, and once again we thank you for your appreciation and insightful feedback!

---

> > ### Comment · Reviewer_rx4i · 2025-04-02
> >
> > I sincerely appreciate the authors' responses. Most of my concerns have been addressed. I also hope the authors can incorporate some theoretical analysis. Considering that this work has some innovation and provides appropriate analysis, I will maintain my original score.

---

> > > ### Author Response · Authors · 2025-04-08
> > >
> > > Dear reviewer rx4i,
> > >
> > > Thank you for your kind reply and for maintaining your positive score for our paper. We are grateful for your interest and support of our work!
> > >
> > > Sincerely,
> > >
> > > Authors

---

### Decision · Program_Chairs · 2025-05-01

**Decision:**

Accept (poster)

**Comment:**

This study examines supervised fine-tuning (SFT) and reinforcement learning (RL), two prominent post-training techniques for foundation models. While reviewers raised valid concerns regarding the model training methodology, theoretical contributions, and novelty of the approach, they unanimously agreed on the importance of investigating how different training strategies affect foundation model performance. After careful consideration, I find that the study's strengths - particularly its timely research question and comprehensive evaluation - ultimately outweigh its limitations. I therefore recommend acceptance.